# NAS-Bench-201: Extending the Scope of Reproducible Neural Architecture Search

**Xuanyi Dong**[†‡] [*] **and Yi Yang**[†]
†ReLER, CAI, University of Technology Sydney, ‡Baidu Research

## Abstract

Neural architecture search (NAS) has achieved breakthrough success in a great number of applications in the past few years. It could be time to take a step back and analyze the good and bad aspects in the field of NAS. A variety of algorithms search architectures under different search space. These searched architectures are trained using different setups, e.g., hyper-parameters, data augmentation, regularization. This raises a comparability problem when comparing the performance of various NAS algorithms. NAS-Bench-101 has shown success to alleviate this problem. In this work, we propose an extension to NAS-Bench-101: *NAS-Bench-201* with a different search space, results on multiple datasets, and more diagnostic information. *NAS-Bench-201* has a fixed search space and provides a unified benchmark for almost any up-to-date NAS algorithms. The design of our search space is inspired from the one used in the most popular cell-based searching algorithms, where a cell is represented as a directed acyclic graph. Each edge here is associated with an operation selected from a predefined operation set. For it to be applicable for all NAS algorithms, the search space defined in *NAS-Bench-201* includes all possible architectures generated by 4 nodes and 5 associated operation options, which results in 15,625 neural cell candidates in total. The training log using the same setup and the performance for each architecture candidate are provided for three datasets. This allows researchers to avoid unnecessary repetitive training for selected architecture and focus solely on the search algorithm itself. The training time saved for every architecture also largely improves the efficiency of most NAS algorithms and brings a more computational cost friendly NAS community for a broader range of researchers. We provide additional diagnostic information such as fine-grained loss and accuracy, which can give inspirations to new designs of NAS algorithms. In further support of the proposed *NAS-Bench-201*, we have analyzed it from many aspects and benchmarked 10 recent NAS algorithms, which verify its applicability.

## 1 Introduction

The deep learning community is undergoing a transition from hand-designed neural architecture (He et al., 2016; Krizhevsky et al., 2012; Szegedy et al., 2015) to automatically designed neural architecture (Zoph & Le, 2017; Pham et al., 2018; Real et al., 2019; Dong & Yang, 2019b; Liu et al., 2019). In its early era, the great success of deep learning was promoted by novel neural architectures, such as ResNet (He et al., 2016), Inception (Szegedy et al., 2015), VGGNet (Simonyan & Zisserman, 2015), and Transformer (Vaswani et al., 2017). However, manually designing one architecture requires human experts to try numerous different operation and connection choices (Zoph & Le, 2017). In contrast to architectures that are manually designed, those automatically found by neural architecture search (NAS) algorithms require much less human interaction and expert effort. These NAS-generated architectures have shown promising results in many domains, such as image recognition (Zoph & Le, 2017; Pham et al., 2018; Real et al., 2019), sequence modeling (Pham et al., 2018; Dong & Yang, 2019b; Liu et al., 2019), etc.

Recently, a variety of NAS algorithms have been increasingly proposed. While these NAS methods are methodically designed and show promising improvements, many setups in their algorithms are

---

[*]Part of this work was done when Xuanyi was a research intern with Baidu Research.

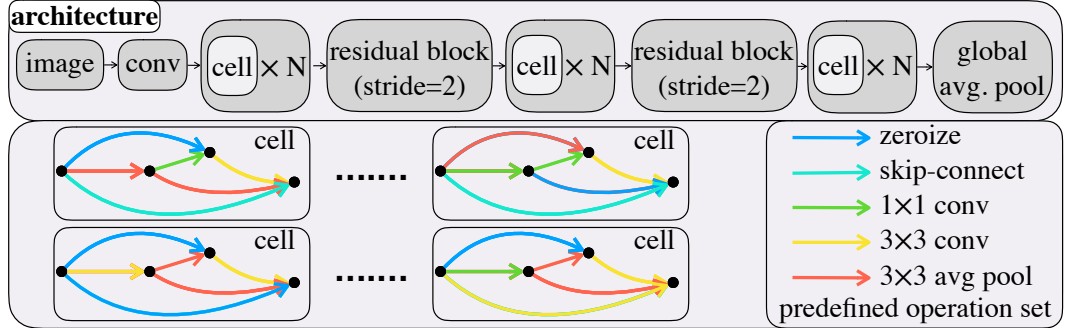

Figure 1: **Top**: the macro skeleton of each architecture candidate. **Bottom-left**: examples of neural cell with 4 nodes. Each cell is a directed acyclic graph, where each edge is associated with an operation selected from a predefined operation set as shown in the **Bottom-right**.

different. (1) Different search space is utilized, e.g., different macro skeletons of the whole architecture (Zoph et al., 2018; Tan et al., 2019) and a different operation set for the micro cell within the skeleton (Pham et al., 2018), etc. (2) After a good architecture is selected, various strategies can be employed to train this architecture and report the performance, e.g., different data augmentation (Ghiasi et al., 2018; Zhang et al., 2018), different regularization (Zoph et al., 2018), different scheduler (Loshchilov & Hutter, 2017), and different selections of hyper-parameters (Liu et al., 2018; Dong & Yang, 2019a). (3) The validation set for testing the performance of the selected architecture is not split in the same way (Liu et al., 2019; Pham et al., 2018). These discrepancies raise a comparability problem when comparing the performance of various NAS algorithms, making it difficult to conclude their contributions.

In response to this problem, NAS-Bench-101 (Ying et al., 2019) and NAS-HPO-Bench (Klein & Hutter, 2019) are proposed. However, some NAS algorithms can not be applied *directly* on NAS-Bench-101, and NAS-HPO-Bench only has 144 candidate architectures, which maybe insufficient to evaluate NAS algorithms. To extend these two benchmarks and towards better reproducibility of NAS methods[1], we propose *NAS-Bench-201* with a fixed cell search space, inspired from the search space used in the most popular neural cell-based searching algorithms (Zoph et al., 2018; Liu et al., 2019). As shown in Figure 1, each architecture consists of a predefined skeleton with a stack of the searched cell. In this way, architecture search is transformed into the problem of searching a good cell. Each cell is represented as a densely-connected directed acyclic graph (DAG) as shown in the bottom section of Figure 1. Here the node represents the sum of the feature maps and each edge is associated with an operation transforming the feature maps from the source node to the target node. The size of the search space is related to the number of nodes defined for the DAG and the size of the operation set. In *NAS-Bench-201*, we choose 4 nodes and 5 representative operation candidates for the operation set, which generates a total search space of 15,625 cells/architectures. Each architecture is trained multiple times on three different datasets. The training log and performance of each architecture are provided for each run. The training accuracy/test accuracy/training loss/test loss after every training epoch for each architecture plus the number of parameters and floating point operations (FLOPs) are accessible.

Hopefully, *NAS-Bench-201* will show its value in the field of NAS research. (1) It provides a unified benchmark for most up-to-date NAS algorithms including all cell-based NAS methods. With *NAS-Bench-201*, researchers can focus on designing robust searching algorithm while avoiding tedious hyper-parameter tuning of the searched architecture. Thus, *NAS-Bench-201* provides a relatively fair benchmark for the comparison of different NAS algorithms. (2) It provides the full training log of each architecture. Unnecessary repetitive training procedure of each selected architecture can be avoided (Liu et al., 2018; Zoph & Le, 2017) so that researchers can target on the essence of NAS, i.e., search algorithm. Another benefit is that the validation time for NAS largely decreases when testing in *NAS-Bench-201*, which provides a computational power friendly environment for more participations in NAS. (3) It provides results of each architecture on multiple datasets. The model transferability can be thoroughly evaluated for most NAS algorithms. (4) In *NAS-Bench-201*, we provide systematic analysis of the proposed search space. We also evaluate 10 recent advanced NAS

---

[1] One parallel concurrent work for the similar purpose is NAS-Bench-1SHOT1 (Zela et al., 2020).

algorithms including reinforcement learning (RL)-based methods, evolutionary strategy (ES)-based methods, differentiable-based methods, etc. We hope our empirical analysis can bring some insights to the future designs of NAS algorithms.

## 2 NAS-Bench-201

Our *NAS-Bench-201* is algorithm-agnostic. Put simply, it is applicable to almost any up-to-date NAS algorithms. In this section, we will briefly introduce our *NAS-Bench-201*. The search space of *NAS-Bench-201* is inspired by cell-based NAS algorithms (Section 2.1). *NAS-Bench-201* evaluates each architecture on three different datasets (Section 2.2). All implementation details of *NAS-Bench-201* are introduced in Section 2.3. *NAS-Bench-201* also provides some diagnostic information which can be used for potentially better designs of future NAS algorithms (discussed in Section 2.4).

### 2.1 ARCHITECTURES IN THE SEARCH SPACE

**Macro Skeleton**. Our search space follows the design of its counterpart as used in the recent neural cell-based NAS algorithms (Liu et al., 2019; Zoph et al., 2018; Pham et al., 2018). As shown in the top of Figure 1, the skeleton is initiated with one 3-by-3 convolution with 16 output channels and a batch normalization layer (Ioffe & Szegedy, 2015). The main body of the skeleton includes three stacks of cells, connected by a residual block. Each cell is stacked $N = 5$ times, with the number of output channels as 16, 32 and 64 for the first, second and third stages, respectively. The intermediate residual block is the basic residual block with a stride of 2 (He et al., 2016), which serves to down-sample the spatial size and double the channels of an input feature map. The shortcut path in this residual block consists of a 2-by-2 average pooling layer with stride of 2 and a 1-by-1 convolution. The skeleton ends up with a global average pooling layer to flatten the feature map into a feature vector. Classification uses a fully connected layer with a softmax layer to transform the feature vector into the final prediction.

**Searched Cell**. Each cell in the search space is represented as a densely connected DAG. The densely connected DAG is obtained by assigning a direction from the $i$-th node to the $j$-th node ($i < j$) for each edge in an undirected complete graph. Each edge in this DAG is associated with an operation transforming the feature map from the source node to the target node. All possible operations are selected from a predefined operation set, as shown in Figure 1(bottom-right). In our *NAS-Bench-201*, the predefined operation set $\mathcal{O}$ has $L = 5$ representative operations: (1) zeroize, (2) skip connection, (3) 1-by-1 convolution, (4) 3-by-3 convolution, and (5) 3-by-3 average pooling layer. The convolution in this operation set is an abbreviation of an operation sequence of ReLU, convolution, and batch normalization. The DAG has $V = 4$ nodes, where each node represents the sum of all feature maps transformed through the associated operations of the edges pointing to this node. We choose $V = 4$ to allow the search space to contain basic residual block-like cells, which requires 4 nodes. Densely connected DAG does not restrict the searched topology of the cell to be densely connected, since we include zeroize in the operation set, which is an operation of dropping the associated edge. Besides, since we do not impose the constraint on the maximum number of edges (Ying et al., 2019), our search space is applicable to most NAS algorithms, including all cell-based NAS algorithms.

### 2.2 DATASETS

We train and evaluate each architecture on CIFAR-10, CIFAR-100 (Krizhevsky et al., 2009), and ImageNet-16-120 (Chrabaszcz et al., 2017). We choose these three datasets because CIFAR and ImageNet (Russakovsky et al., 2015) are the most popular image classification datasets.

We split each dataset into training, validation and test sets to provide a consistent training and evaluation settings for previous NAS algorithms (Liu et al., 2019). Most NAS methods use the validation set to evaluate architectures after the architecture is optimized on the training set. The validation performance of the architectures serves as supervision signals to update the searching algorithm. The test set is to evaluate the performance of each searching algorithm by comparing the indicators (e.g., accuracy, model size, speed) of their selected architectures. Previous methods use different splitting strategies, which may result in various searching costs and unfair comparisons. We hope to use the proposed splits to unify the training, validation and test sets for a fairer comparison.

**CIFAR-10**: It is a standard image classification dataset and consists of 60K 32×32 colour images in 10 classes. The original training set contains 50K images, with 5K images per class. The original test set contains 10K images, with 1K images per class. Due to the need of validation set, we split all 50K training images in CIFAR-10 into two groups. Each group contains 25K images with 10 classes. We regard the first group as the new training set and the second group as the validation set.

**CIFAR-100**: This dataset is just like CIFAR-10. It has the same images as CIFAR-10 but categorizes each image into 100 fine-grained classes. The original training set on CIFAR-100 has 50K images, and the original test set has 10K images. We randomly split the original test set into two group of equal size — 5K images per group. One group is regarded as the validation set, and another one is regarded as the new test set.

**ImageNet-16-120**: We build ImageNet-16-120 from the down-sampled variant of ImageNet (ImageNet16×16). As indicated in Chrabaszcz et al. (2017), down-sampling images in ImageNet can largely reduce the computation costs for optimal hyper-parameters of some classical models while maintaining similar searching results. Chrabaszcz et al. (2017) down-sampled the original ImageNet to 16×16 pixels to form ImageNet16×16, from which we select all images with label $\in [1, 120]$ to construct ImageNet-16-120. In sum, ImageNet-16-120 contains 151.7K training images, 3K validation images, and 3K test images with 120 classes.

By default, in this paper, "the training set", "the validation set", "the test set" indicate the new training, validation, and test sets, respectively.

## 2.3 ARCHITECTURE PERFORMANCE

**Training Architectures.** In order to unify the performance of every architecture, we give the performance of every architecture in our search space. In our *NAS-Bench-201*, we follow previous literature to set up the hyper-parameters and training strategies (Zoph et al., 2018; Loshchilov & Hutter, 2017; He et al., 2016). We train each architecture with the same strategy, which is shown in Table 1. For simplification, we denote all hyper-parameters for training a model as a set $\mathcal{H}$, and we use $\mathcal{H}^\dagger$ to denote the values of hyper-parameter that we use. Specifically, we train each architecture via Nesterov momentum SGD, using the cross-entropy loss for 200 epochs in total. We set the weight de-

Table 1: The training hyper-parameter set $\mathcal{H}^\dagger$.

| optimizer | SGD | initial LR | 0.1 |
|---|---|---|---|
| Nesterov | ✓ | ending LR | 0 |
| momentum | 0.9 | LR schedule | cosine |
| weight decay | 0.0005 | epoch | 200 |
| batch size | 256 | initial channel | 16 |
| $V$ | 4 | $N$ | 5 |
| random flip | p=0.5 | random crop | ✓ |
| normalization | ✓ | | |

cay as 0.0005 and decay the learning rate from 0.1 to 0 with a cosine annealing (Loshchilov & Hutter, 2017). We use the same $\mathcal{H}^\dagger$ on different datasets, except for the data augmentation which is slightly different due to the image resolution. On CIFAR, we use the random flip with probability of 0.5, the random crop 32×32 patch with 4 pixels padding on each border, and the normalization over RGB channels. On ImageNet-16-120, we use a similar strategy but random crop 16×16 patch with 2 pixels padding on each border. Apart from using $\mathcal{H}^\dagger$ for all datasets, we also use a different hyper-parameter set $\mathcal{H}^\ddagger$ for CIFAR-10. It is similar to $\mathcal{H}^\dagger$ but its total number of training epochs is 12. In this way, we could provide bandit-based algorithms (Falkner et al., 2018; Li et al., 2018) more options for the usage of short training budget (see more details in appendix).

**Metrics**. We train each architecture with different random seeds on different datasets. We evaluate each architecture $A$ after every training epoch. *NAS-Bench-201* provides the training, validation, and test loss as well as accuracy. We show the supported metrics on different datasets in Table 2. Users can easily use our API to query the results of each trial of $A$, which has negligible computational costs. In this way, researchers could significantly speed up their searching algorithm on these datasets and focus solely on the essence of NAS.

Table 2: *NAS-Bench-201* provides the following metrics with $\mathcal{H}^\dagger$. 'Acc.' means accuracy.

| Dataset | Train Loss/Acc. | Eval Loss/Acc. |
|---|---|---|
| CIFAR-10 | train set | valid set |
| CIFAR-10 | train+valid set | test set |
| CIFAR-100 | train set | valid set |
| CIFAR-100 | train set | test set |
| ImageNet-16-120 | train set | valid set |
| ImageNet-16-120 | train set | test set |

We list the training/test loss/accuracies over different split sets on four datasets in Table 2. On CIFAR-10, we train the model on the training set and evaluate it on the validation set. We also train the model on the training and validation set and

| | #archit -ectures | #data -sets | $|\mathcal{O}|$ | search space constraint | Supported NAS algorithms | | | | Diagnostic information |
| --- | --- | --- | --- | --- | --- | --- | --- | --- | --- |
| | | | | | RL | ES | Diff. | HPO | |
| NAS-Bench-101 | 510M | 1 | 3 | constrain #edges | partial | partial | none | most | – |
| *NAS-Bench-201* | 15.6K | 3 | 5 | no constraint | all | all | all | most | fine-grained info., param., etc |

Table 3: We summarize some characteristics of NAS-Bench-101 and *NAS-Bench-201*. Our *NAS-Bench-201* can **directly** be applicable to almost any up-to-date NAS algorithms. In contrast, as pointed in (Ying et al., 2019), NAS algorithms based on parameter sharing or network morphisms cannot be **directly** evaluated on NAS-Bench-101. Besides, *NAS-Bench-201* provides train/validation/test performance on three (one for NAS-Bench-101) different datasets so that the generality of NAS algorithms can be evaluated. It also provides some diagnostic information that may provide insights to design better NAS algorithms.

evaluate it on the test set. These two paradigm follow the typical experimental setup on CIFAR-10 in previous literature (Liu et al., 2018; Zoph et al., 2018; Liu et al., 2018; Pham et al., 2018). On CIFAR-100 and ImageNet-16-120, we train the model on the training set and evaluate it on both validation and test sets.

## 2.4 Diagnostic Information

Validation accuracy is a commonly used supervision signal for NAS. However, considering the expensive computational costs for evaluating the architecture, the signal is too sparse. In our *NAS-Bench-201*, we also provide some diagnostic information which is some extra statistics obtained during training each architecture. Collecting these statistics almost involves no extra computation cost but may provide insights for better designs and training strategies of different NAS algorithms, such as platform-aware NAS (Tan et al., 2019), accuracy prediction (Baker et al., 2018), mutation-based NAS (Cai et al., 2018; Chen et al., 2016), etc.

**Architecture Computational Costs:** *NAS-Bench-201* provides three computation metrics for each architecture — the number of parameters, FLOPs, and latency. Algorithms that target on searching architectures with computational constraints, such as models on edge devices, can use these metrics directly in their algorithm designs without extra calculations.

**Fine-grained training and evaluation information.** *NAS-Bench-201* tracks the changes in loss and accuracy of every architecture after every training epochs. These fine-grained training and evaluation information shows the tendency of the architecture performance and could indicate some attributes of the model, such as the speed of convergence, the stability, the over-fitting or under-fitting levels, etc. These attributes may benefit the designs of NAS algorithms. Besides, some methods learn to predict the final accuracy of an architecture based on the results of few early training epochs (Baker et al., 2018). These algorithm can be trained faster and the performance of the accuracy prediction can be evaluated using the fine-grained evaluation information.

**Parameters of optimized architecture.** Our *NAS-Bench-201* releases the trained parameters for each architecture. This can provide ground truth label for hypernetwork-based NAS methods (Zhang et al., 2019; Brock et al., 2018), which learn to generate parameters of an architecture. Other methods mutate an architecture to become another one (Real et al., 2019; Cai et al., 2018). With *NAS-Bench-201*, researchers could directly use the off-the-shelf parameters instead of training from scratch and analyze how to transfer parameters from one architecture to another.

## 3 Difference with existing NAS benchmarks

To the best of our knowledge, NAS-Bench-101 (Ying et al., 2019) is the only existing large-scale architecture dataset. Similar to *NAS-Bench-201*, NAS-Bench-101 also transforms the problem of architecture search into the problem of searching neural cells, represented as a DAG. Differently, NAS-Bench-101 defines operation candidates on the node, whereas we associate operations on the edge as inspired from (Liu et al., 2019; Dong & Yang, 2019b; Zoph et al., 2018). We summarize characteristics of our *NAS-Bench-201* and NAS-Bench-101 in Table 3. The main highlights of our *NAS-Bench-201* are as follows. (1) *NAS-Bench-201* is algorithm-agnostic while NAS-Bench-

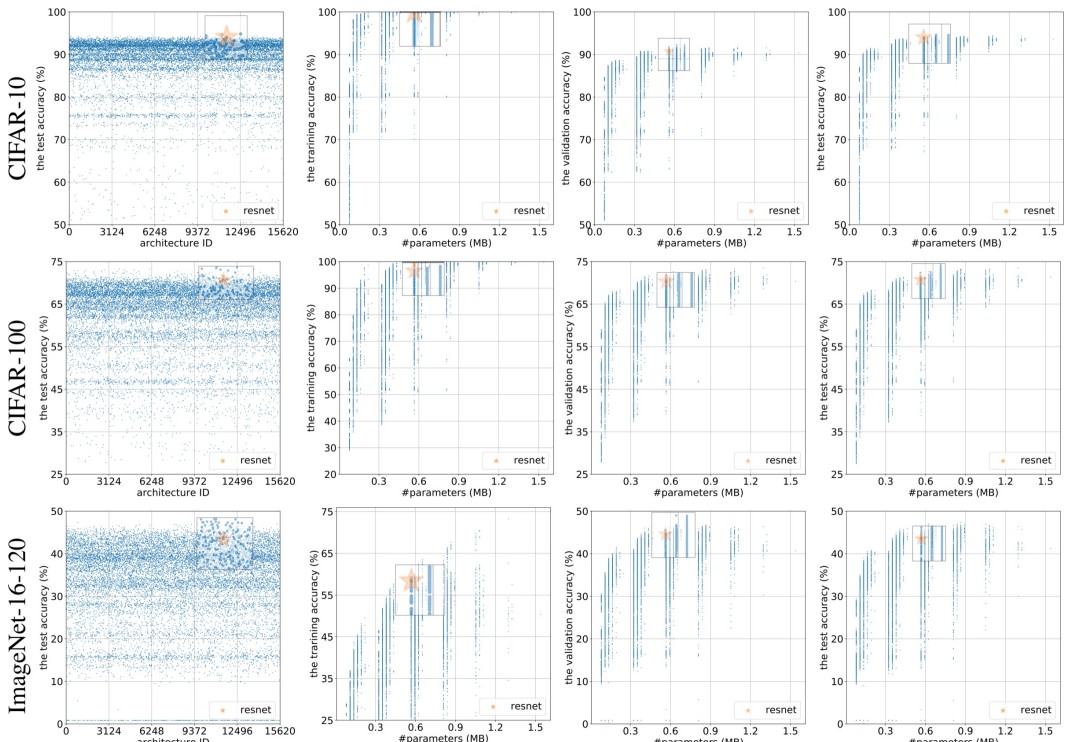

Figure 2: Training, validation, test accuracy of each architecture on CIFAR-10, CIFAR-100, and ImageNet-16-120. We also visualize the results of ResNet in the orange star marker.

101 without any modification is only applicable to selected algorithms (Yu et al., 2020; Zela et al., 2020). The original complete search space, based on the nodes in NAS-Bench-101, is extremely huge. So, it is exceedingly difficult to efficiently traverse the training of all architectures. To trade off the computational cost and the size of the search space, they constrain the maximum number of edges in the DAG. However, it is difficult to incorporate this constraint in all NAS algorithms, such as NAS algorithms based on parameter-sharing (Liu et al., 2019; Pham et al., 2018). Therefore, many NAS algorithms cannot be directly evaluated on NAS-Bench-101. Our *NAS-Bench-201* solves this problem by sacrificing the number of nodes and including all possible edges so that our search space is algorithm-agnostic. (2) We provide extra diagnostic information, such as architecture computational cost, fine-grained training and evaluation time, etc., which give inspirations to better and efficient designs of NAS algorithms utilizing these diagnostic information.

NAS-HPO-Bench (Klein & Hutter, 2019) evaluated 62208 configurations in the joint NAS and hyper-parameter space for a simple 2-layer feed-forward network. Since NAS-HPO-Bench has only 144 architectures, it could be insufficient to evaluate different NAS algorithms.

## 4 ANALYSIS OF *NAS-Bench-201*

**An overview of architecture performance.** The performance of each architecture is shown in Figure 2. We show the test accuracy of every architecture in our search space in the left column of Figure 2. The training, validation and test accuracy with respect to the number of parameters are shown in the rest three columns, respectively. Results show that a different number of parameters will affect the performance of the architectures, which indicates that the choices of operations are essential in NAS. We also observe that the performance of the

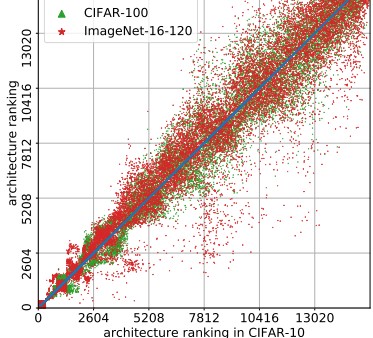

Figure 3: The ranking of each architecture on three datasets, sorted by the ranking in CIFAR-10.

architecture can vary even when the number of parameters stays the same. This observation indicates the importance of how the operations/cells are connected. We compare the architectures with a clas-

sical human-designed architecture (ResNet) in all cases, which is indicated by an orange star mark. ResNet shows competitive performance in three datasets, however, it still has room to improve, i.e., about 2% compared to the best architecture in CIFAR-100 and ImageNet-16-120, about 1% compared to the best one with the same amount of parameters in CIFAR-100 and ImageNet-16-120.

**Architecture ranking on three datasets.** The ranking of every architecture in our search space is shown in Figure 3, where the architecture ranked in CIFAR-10 (x-axis) is ranked as in y-axis in CIFAR-100 and ImageNet-16-120, indicated by green and red markers respectively. The performance of the architectures shows a generally consistent ranking over the three datasets with slightly different variance, which serves to test the generality of the searching algorithm.

**Correlations of validation and test accuracies.** We visualize the correlation between the validation and test accuracy within one dataset and across datasets in Figure 4. The correlation within one dataset is high compared to cross-dataset correlation. The correlation dramatically decreases as we only pick the top performing architectures. When we directly transfer the best architecture in one dataset to another (a vanilla strategy), it can not 100% secure a good performance. This phenomena is a call for better transferable NAS algorithms instead of vanilla strategy.

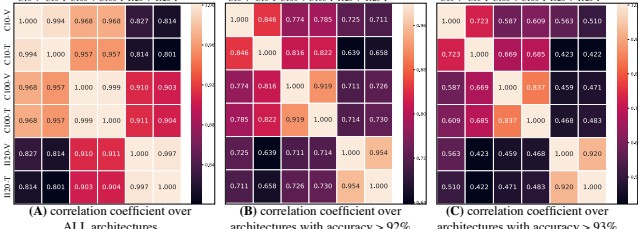

**(A)** correlation coefficient over ALL architectures    **(B)** correlation coefficient over architectures with accuracy > 92%    **(C)** correlation coefficient over architectures with accuracy > 93%

**Dynamic ranking of architectures.** We show the ranking of the performance of all architectures in different time stamps in Figure 5. The ranking based on the validation set (y axis) gradually converges to the ranking based on the final test accuracy (x axis).

Figure 4: We report the correlation coefficient between the accuracy on 6 sets, i.e., CIFAR-10 validation set (C10-V), CIFAR-10 test set (C10-T), CIFAR-100 validation set (C100-V), CIFAR-100 test set (C100-T), ImageNet-16-120 validation set (I120-V), ImageNet-16-120 test set (I120-T).

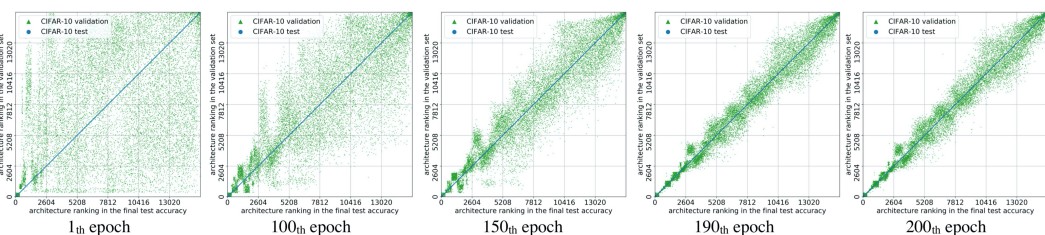

| 1th epoch | 100th epoch | 150th epoch | 190th epoch | 200th epoch |

Figure 5: The ranking of all architectures based on the validation accuracy at different time stamps (y axis) sorted by the final test accuracy (x axis).

## 5 BENCHMARK

In this section, we evaluate 10 recent searching methods on our *NAS-Bench-201*, which can serve as baselines for future NAS algorithms in our dataset. Specifically, we evaluate some typical NAS algorithms: (I) Random Search algorithms, e.g., random search (RS) (Bergstra & Bengio, 2012), random search with parameter sharing (RSPS) (Li & Talwalkar, 2019). (II) ES methods, e.g., REA (Real et al., 2019). (III) RL algorithms, e.g., REINFORCE (Williams, 1992), ENAS (Pham et al., 2018). (IV) Differentiable algorithms. e.g., first order DARTS (DARTS-V1) (Liu et al., 2019), second order DARTS (DARTS-V2), GDAS (Dong & Yang, 2019b), and SETN (Dong & Yang, 2019a). (V) HPO methods, e.g., BOHB (Falkner et al., 2018). We experimented all NAS algorithms on a single GeForce GTX 1080 Ti GPU.

| accelerate | RS | RSPS | DARTS-V1 | DARTS-V2 | GDAS | SETN | REA | REINFORCE | ENAS | BOHB |
|---|---|---|---|---|---|---|---|---|---|---|
| search | ✓ | ✗ | ✗ | ✗ | ✗ | ✗ | ✓ | ✓ | ✓ | ✓ |
| evaluation | ✓ | ✓ | ✓ | ✓ | ✓ | ✓ | ✓ | ✓ | ✓ | ✓ |

Table 4: The utility of our *NAS-Bench-201* for different NAS algorithms. We show whether a NAS algorithm can use our *NAS-Bench-201* to accelerate the searching and evaluation procedure.

| Method | Search (seconds) | CIFAR-10 | | CIFAR-100 | | ImageNet-16-120 | |
|---|---|---|---|---|---|---|---|
| | | validation | test | validation | test | validation | test |
| RSPS | 8007.13 | 80.42±3.58 | 84.07±3.61 | 52.12±5.55 | 52.31±5.77 | 27.22±3.24 | 26.28±3.09 |
| DARTS-V1 | 11625.77 | 39.77±0.00 | 54.30±0.00 | 15.03±0.00 | 15.61±0.00 | 16.43±0.00 | 16.32±0.00 |
| DARTS-V2 | 35781.80 | 39.77±0.00 | 54.30±0.00 | 15.03±0.00 | 15.61±0.00 | 16.43±0.00 | 16.32±0.00 |
| GDAS | 31609.80 | 89.89±0.08 | 93.61±0.09 | 71.34±0.04 | 70.70±0.30 | 41.59±1.33 | 41.71±0.98 |
| SETN | 34139.53 | 84.04±0.28 | 87.64±0.00 | 58.86±0.06 | 59.05±0.24 | 33.06±0.02 | 32.52±0.21 |
| ENAS | 14058.80 | 37.51±3.19 | 53.89±0.58 | 13.37±2.35 | 13.96±2.33 | 15.06±1.95 | 14.84±2.10 |
| RSPS[†] | 7587.12 | 84.16±1.69 | 87.66±1.69 | 59.00±4.60 | 58.33±4.34 | 31.56±3.28 | 31.14±3.88 |
| DARTS-V1[†] | 10889.87 | 39.77±0.00 | 54.30±0.00 | 15.03±0.00 | 15.61±0.00 | 16.43±0.00 | 16.32±0.00 |
| DARTS-V2[†] | 29901.67 | 39.77±0.00 | 54.30±0.00 | 15.03±0.00 | 15.61±0.00 | 16.43±0.00 | 16.32±0.00 |
| GDAS[†] | 28925.91 | 90.00±0.21 | 93.51±0.13 | 71.14±0.27 | 70.61±0.26 | 41.70±1.26 | 41.84±0.90 |
| SETN[†] | 31009.81 | 82.25±5.17 | 86.19±4.63 | 56.86±7.59 | 56.87±7.77 | 32.54±3.63 | 31.90±4.07 |
| ENAS[†] | 13314.51 | 39.77±0.00 | 54.30±0.00 | 15.03±0.00 | 15.61±0.00 | 16.43±0.00 | 16.32±0.00 |
| REA | 0.02 | 91.19±0.31 | 93.92±0.30 | 71.81±1.12 | 71.84±0.99 | 45.15±0.89 | 45.54±1.03 |
| RS | 0.01 | 90.93±0.36 | 93.70±0.36 | 70.93±1.09 | 71.04±1.07 | 44.45±1.10 | 44.57±1.25 |
| REINFORCE | 0.12 | 91.09±0.37 | 93.85±0.37 | 71.61±1.12 | 71.71±1.09 | 45.05±1.02 | 45.24±1.18 |
| BOHB | 3.59 | 90.82±0.53 | 93.61±0.52 | 70.74±1.29 | 70.85±1.28 | 44.26±1.36 | 44.42±1.49 |
| ResNet | N/A | 90.83 | 93.97 | 70.42 | 70.86 | 44.53 | 43.63 |
| **optimal** | | 91.61 | 94.37 | 73.49 | 73.51 | 46.77 | 47.31 |

Table 5: We evaluate *10* different searching algorithms in our *NAS-Bench-201*. The first block shows results of parameter sharing based NAS methods. The second block is similar to the first one, however, BN layers in the searching cells do not keep running estimates but always use batch statistics. The third block shows results of NAS methods without parameter sharing. Each algorithm uses the training and validation set of CIFAR-10 for searching. We show results of their searched architectures for (1) training on the CIFAR-10 train set and evaluating on its validation set; (2) training on the CIFAR-10 train+validation sets and evaluating on its test set; (3) training on the CIFAR-10 or ImageNet-16-120 train set and evaluating on their validation or test sets. "optimal" indicates the highest mean accuracy for each set. We report the mean and std of 500 runs for RS, REA, REINFORCE, and BOHB and of 3 runs for RSPS, DARTS, GDAS, SETN, and ENAS.

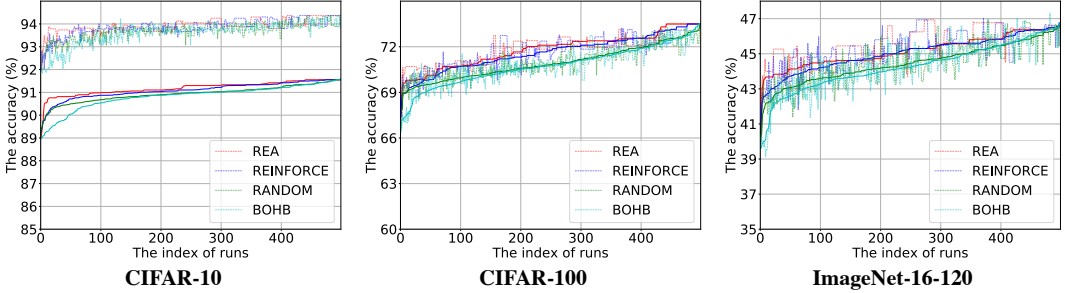

**CIFAR-10**  **CIFAR-100**  **ImageNet-16-120**

Figure 6: We show results of 500 runs for RS, REA, REINFORCE, and BOHB on CIFAR-10. The architecture is searched on CIFAR-10 and we report its validation accuracy (*solid line*) and test accuracy (*dashed line*) on three datasets. Each individual run is sorted by the validation accuracy of the searched architecture.

We show the benefits for speed using our *NAS-Bench-201* for different NAS algorithms in Table 4. For each NAS algorithm, once the searching procedure finished and the final architecture is found, our *NAS-Bench-201* can directly return the performance of this architecture. With *NAS-Bench-201*, NAS algorithms without parameter sharing can significantly reduce the searching time into seconds. Notably, it still requires several GPU hours for NAS algorithms with parameter sharing to complete the searching.

All algorithms use the training and validation set of CIFAR-10 to search architectures. In Table 5, Figure 6, Figure 7, and Figure 8, we report the performance of the searched architectures plus the optimal architecture on three datasets. We make the following observations: (1) NAS methods without parameter sharing (REA, RS, REINFORCE, and BOHB) outperform others. This be because training a model for a few epochs with the converged LR scheduler ($\mathcal{H}^{\ddagger}$) can provide a good relative

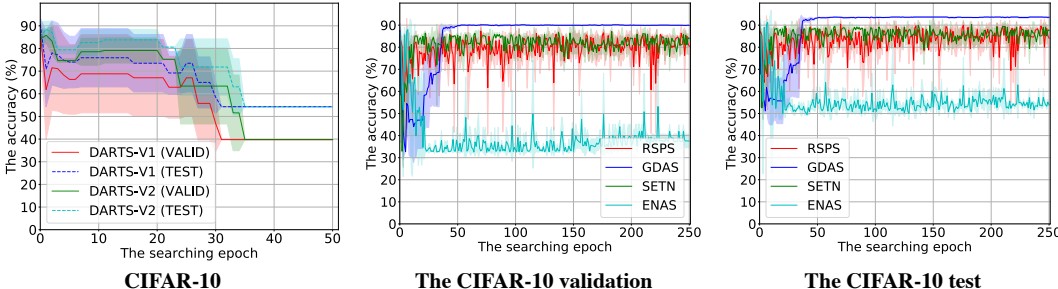

Figure 7: *Results keeping keep running estimates for BN layers in each searching cell*. We use parameter sharing based NAS methods to search the architecture on CIFAR-10. After each searching epoch, we derive the architecture and show its validation accuracy (VALID) and test accuracy (TEST) on CIFAR-10. The 0-th epoch indicates the architecture is derived from the randomly initialized architecture encoding.

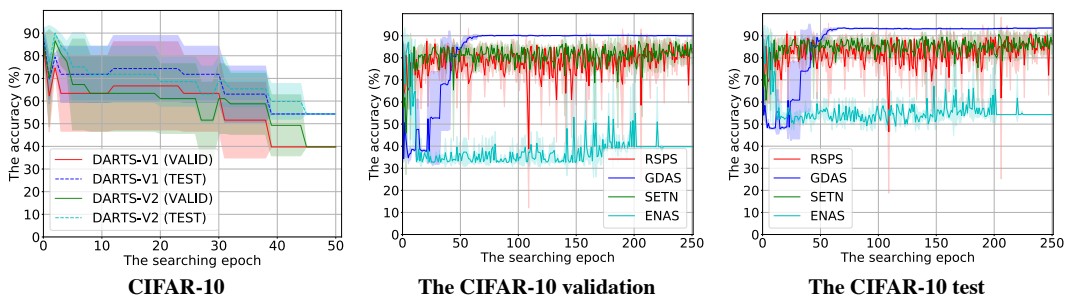

Figure 8: *Results using batch statistics without keeping keep running estimates for BN layers in each searching cell*. We use parameter sharing based NAS methods to search the architecture on CIFAR-10. After each searching epoch, we derive the architecture and show its validation accuracy (VALID) and test accuracy (TEST) on CIFAR-10. The 0-th epoch indicates the architecture is derived from the randomly initialized architecture encoding.

ranking of each architecture. (2) DARTS-V1 and DARTS-V2 quickly converge to find the architecture whose edges are all skip connection. A possible reason is that the original hyper-parameters of DARTS are chosen for their search space instead of ours. (3) The strategy of BN layers can significantly effect the NAS methods with parameter sharing. Using batch statistics are better than keep running estimates of the mean and variance. (4) Using our fine-grained information, REA, REINFORCE and RS can be finished in seconds which could significantly reduce the search costs and let researchers focus solely on the search algorithm itself.

In Figure 7 and Figure 8, we show the performance of the architecture derived from each algorithm per searching epoch. DARTS-V1 will gradually over-fit to an architecture with all skip-connection operations. DARTS-V2 can alleviate this problem to some extent but will still over-fit after more epochs. It can further alleviate this problem by using batch statistics for BN layers. We train RSPS, GDAS, SETN, and ENAS five times longer than DARTS (250 epochs vs. 50 epochs). This is because at every iteration, RSPS, GDAS, SETN, and ENAS only optimize $\frac{1}{|\mathcal{O}|=5}$ parameters of the shared parameters, whereas DARTS optimize all shared parameters. The searched architecture performs similar for GDAS after 50 searching epochs. RSPS and SETN show a higher variance of the searched architecture compared to GDAS.

**Clarification.** We have tried our best to implement each method. However, still, some algorithms might obtain non-optimal results since their hyper-parameters might not fit our *NAS-Bench-201*. We empirically found that some NAS algorithms are sensitive to some hyper-parameters, whereas we try to compare them in a fair way as we can (Please see more explanation in Appendix). If researchers can provide better results with different hyper-parameters, we are happy to update results according to the new experimental results. We also welcome more NAS algorithms to test on our dataset and would include them accordingly.

## 6    DISCUSSION

**How to avoid over-fitting on *NAS-Bench-201*?**    Our *NAS-Bench-201* provides a benchmark for NAS algorithms, aiming to provide a fair and computational cost-friendly environment to the NAS community. The trained architecture and the easy-to-access performance of each architecture might provide some insidious ways for designing algorithms to over-fit the best architecture in our *NAS-Bench-201*. Thus, we propose some rules which we wish the users will follow to achieve the original intention of *NAS-Bench-201*, a fair and efficient benchmark.

*1. No regularization for a specific operation.* Since the best architecture is known in our benchmark, specific designs to fit the structural attributes of the best performed architecture are insidious ways to fit our *NAS-Bench-201*. For example, as mentioned in Section 5, we found that the best architecture with the same amount of parameters for CIFAR10 on *NAS-Bench-201* is ResNet. Restrictions on the number of residual connections is a way to over-fit the CIFAR10 benchmark. While this can give a good result on this benchmark, the searching algorithm might not generalize to other benchmarks.

*2. Use the provided performance.* The training strategy affects the performance of the architecture. We suggest the users stick to the performance provided in our benchmark even if it is feasible to use other $\mathcal{H}$ to get a better performance. This provides a fair comparison with other algorithms.

*3. Report results of multiple searching runs.* Since our benchmark can help to largely decrease the computational cost for a number of algorithms. Multiple searching runs give stable results of the searching algorithm with acceptable time cost.

**Limitation regarding to hyper-parameter optimization (HPO).** The performance of an architecture depends on the hyper-parameters $\mathcal{H}$ for its training and the optimal configuration of $\mathcal{H}$ may vary for different architectures. In *NAS-Bench-201*, we use the same configuration for all architectures, which may bring biases to the performance of some architectures. One related solution is HPO, which aims to search the optimal hyper-parameter configuration. However, searching the optimal hyper-parameter configurations and the architecture in one shot is too computationally expensive and still is an open problem.

**Potential designs using diagnostic information in *NAS-Bench-201*.** As pointed in Section 2.4, different kinds of diagnostic information are provided. We hope that more insights about NAS could be found by analyzing these diagnostic information and further motivate potential solutions for NAS. For example, parameter sharing (Pham et al., 2018) is the crucial technique to improve the searching efficiency, but the shared parameter would sacrifice the accuracy of each architecture. Could we find a better way to share parameters of each architecture from the learned 15,625 models' parameters?

**Generalization ability of the search space**. It is important to test the generalization of observations on this dataset. An idea strategy is to do all benchmark experiments on a much larger search space. Unfortunately, it is prohibitive regarding the expensive computational cost. We bring some results from (Ying et al., 2019) and (Zela et al., 2020) to provide some preliminary evidence of generalization. In Figure 2, we show the rankings of RS, REA, and REINFORCE is ( REA > REINFORCE > RS ). This is consistent with results in NAS-Bench-101, which contains more architecture candidates. For NAS methods with parameter sharing, we find that GDAS $\geq$ DARTS $\geq$ ENAS, which is also consistent with results in NAS-Bench-1SHOT1. Therefore, observations from our *NAS-Bench-201* may generalize to other search spaces.

## 7    CONCLUSION & FUTURE WORK

In this paper, we introduce *NAS-Bench-201* that extends the scope of reproducible NAS. In *NAS-Bench-201*, almost any NAS algorithms can be *directly* evaluated. We train and evaluate 15,625 architecture on three different datasets, and we provide results regarding different metrics. We comprehensively analyze our dataset and test some recent NAS algorithms on *NAS-Bench-201* to serve as baselines for future works. In future, we will (1) consider HPO and NAS together and (2) much larger search space. We welcome researchers to try their NAS algorithms on our *NAS-Bench-201* and would update the paper to include their results.

**Acknowledgements.** We thank the ICLR area chair, ICLR reviewers, and authors of NAS-Bench-101 for the constructive suggestions during the rebuttal and revision period.

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

| EPOCHS | TOTAL | CIFAR-10 | | CIFAR-100 | | ImageNet-16-120 | |
|---|---|---|---|---|---|---|---|
| | | validation | test | validation | test | validation | test |
| 6 | 12 ($\mathcal{H}^{\ddagger}$) | 0.7767 | 0.7627 | 0.8086 | 0.8095 | 0.8052 | 0.7941 |
| 12 | 12 ($\mathcal{H}^{\ddagger}$) | 0.9110 | 0.8983 | 0.9361 | 0.9368 | 0.9062 | 0.8952 |
| 12 | 200 ($\mathcal{H}^{\dagger}$) | 0.7520 | 0.7396 | 0.8071 | 0.8080 | 0.8167 | 0.8092 |
| 24 | 200 ($\mathcal{H}^{\dagger}$) | 0.7705 | 0.7594 | 0.8280 | 0.8290 | 0.8286 | 0.8217 |
| 100 | 200 ($\mathcal{H}^{\dagger}$) | 0.7938 | 0.7900 | 0.8529 | 0.8540 | 0.8262 | 0.8211 |
| 150 | 200 ($\mathcal{H}^{\dagger}$) | 0.8955 | 0.8926 | 0.9239 | 0.9246 | 0.8506 | 0.8425 |
| 175 | 200 ($\mathcal{H}^{\dagger}$) | 0.9834 | 0.9782 | 0.9743 | 0.9744 | 0.8539 | 0.8423 |
| 200 | 200 ($\mathcal{H}^{\dagger}$) | 0.9993 | 0.9937 | 0.9672 | 0.9671 | 0.8259 | 0.8124 |

Table 6: We compare the correlation of different training strategies. The correlation coefficient between the validation accuracy after several training epochs on CIFAR-10 and (1) the validation accuracy of full trained models on the CIFAR-10 training set, (2) the test accuracy on CIFAR-10 trained with the training and validation sets, (3) the validation/test accuracy on CIFAR-100 trained with the CIFAR-100 training set, (4) the validation/test accuracy on ImageNet-16-120 trained with the ImageNet-16-120 training set. We use the validation accuracy after "EPOCHS" training epochs, where the the cosine annealing converged after "TOTAL" epochs.

## A   MORE DETAILS OF *NAS-Bench-201*

**Number of unique architectures.** In our *NAS-Bench-201*, we encode each architecture by a 6-dimensional vector. The $i$-th value in this vector indicates the operation in the $i$-th edge in a cell. Since we have 5 possible operations, there are $5^6 = 15625$ total unique models in this encoding. If we identify the isomorphic cell caused by the "skip-connect" operation, there are 12751 unique topology structures. If we identify the isomorphic cell caused by both "skip-connect" and "zeroize" operations, there are only 6466 unique topology structures. Note that, due to the numerical error, when given the same inputs, two architectures with the isomorphic cell might have different outputs.

Note that, when we build our *NAS-Bench-201*, we train and evaluate every architecture without considering isomorphism.

***NAS-Bench-201* with bandit-based algorithms.** Bandit-based algorithms, such as Hyperband (Li et al., 2018) and BOHB (Falkner et al., 2018), usually train models with a short time budget. In our *NAS-Bench-201*, on CIFAR-10, we provide two options if you want to obtain the performance of a model trained with a short time budget: (1) Results from $\mathcal{H}^{\ddagger}$, where the cosine annealing converged at the 12-th epoch. (2) Results from $\mathcal{H}^{\dagger}$, where the cosine annealing converged at the 200-th epoch. As shown in Table 6, the performance of these converged networks is much more likely to correlate highly with the performance after a larger number of iterations than just taking an earlier point of a single cosine annealing trajectory. Therefore, we choose the first option for all NAS algorithms that do not use parameter sharing.

## B   IMPLEMENTATION DETAILS

Based on the publicly available codes, we re-implement 10 NAS algorithms by ourselves to search architectures on our *NAS-Bench-201*. We provide the implementation details of each searching algorithm below.

We consider the searching time of the first order DARTS as a baseline (about 12000 seconds on CIFAR-10). When evaluating RS, REINFORCE, ENAS, and BOHB, we set the total time budget as 12000 seconds for them. By default, for NAS algorithms with parameter sharing, we follow

most hyper-parameters from DARTS and do not learn the scale and shift parameters for BN layers in each searching cell. We setup the searching procedure of RSPS, GDAS, SETN, ENAS five times longer than DARTS, because they optimize $\frac{1}{5}$ of parameters but DARTS optimize all parameters per iteration. Most configurations can be found at `https://github.com/D-X-Y/AutoDL-Projects/tree/master/configs/nas-benchmark/algos`.

**Random search (RS)** (Bergstra & Bengio, 2012). We randomly select architectures until the total training time plus the time of one evaluation procedure reaches the total budget. We use the validation accuracy after 12 training epochs ($\mathcal{H}^{\ddagger}$), which can be obtained directly in our *NAS-Bench-201* as discussed in Section 2.4. The architecture with the highest validation accuracy is selected as the final searched architecture.

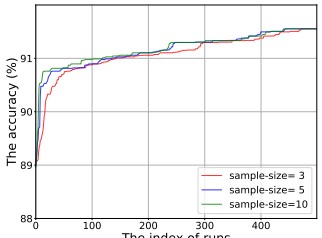

Figure 9: The effect of different sample sizes for REA on the CIFAR-10 validation set.

**Regularized evolution for image classifier architecture search (REA)** (Real et al., 2019). We set the initial population size as 10, the number of cycles as infinity. The sample size is chosen as 10 from [3, 5, 10], according to Figure 9. We finish the algorithm once the simulated training time of the traversed architecture reaches the time budgets (12000 seconds). We use the validation accuracy after 12 training epochs ($\mathcal{H}^{\ddagger}$) as the fitness.

**REINFORCE** (Williams, 1992). We follow (Ying et al., 2019) to use the REINFORCE algorithm as a baseline RL method. We use an architecture encoding to parameterize each candidate in our search space as (Liu et al., 2019; Dong & Yang, 2019b). We use the validation accuracy after 12 training epochs ($\mathcal{H}^{\ddagger}$ as the reward in REINFORCE. The architecture encoding is optimized via Adam. We evaluate the learning rate from [0.01, 0.02, 0.05, 0.1, 0.2, 0.5] following (Ying et al., 2019). According to Figure 10, the learning date is set as . The momentum for exponential moving average of 0.9. We finish the training once the simulated training time reaches the time budgets (12000 seconds).

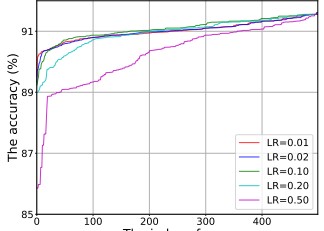

Figure 10: We evaluate the effect of different learning rates for REINFORCE, and report the CIFAR-10 validation accuracy of the searched architecture.

**The first order and second order DARTS (DARTS-V1 and DARTS-V2)** (Liu et al., 2019). We train the shared parameters via Nesterov momentum SGD, using the cross-entropy loss for 50 epochs in total. We set weight decay as 0.0005 and momentum of 0.9. We decay the learning rate from 0.025 to 0.001 via cosine learning rate scheduler and clip the gradient by 5. We train the architecture encoding via Adam with the learning rate of 0.0003 and the weight decay of 0.001. We use the batch size of 64. The random horizontal flipping, random cropping with padding, and normalization are used for data augmentation. We choose these hyper-parameters following (Liu et al., 2019).

**Random search with parameter sharing (RSPS)** (Li & Talwalkar, 2019). We train RSPS with the similar hyper-parameters as that of DARTS. Differently, we train the algorithm in 250 epochs in total. During each searching iteration, we randomly sample one architecture in each batch training. Each architecture uses the training mode for BN during training and the evaluation mode during evaluation (Paszke et al., 2017). After training the shared parameters, we evaluate 100 randomly selected architectures with the shared parameters. For each architecture, we randomly choose one mini-batch with 256 validation samples to estimate the validation accuracy instead of using the whole validation set to calculate the precise validation accuracy. The one with the highest estimated validation accuracy will be selected. With the size of this mini-batch increasing, the more precise validation accuracy would be obtained and the better architecture would be selected. However, the searching costs will also be increased. We use the size of 256 to trade-off the accuracy and cost.

**Gradient-based search using differentiable architecture sampler (GDAS)** (Dong & Yang, 2019b). We use the most hyper-parameters as that of DARTS but train it for 250 epochs in total. The Gumbel-Softmax temperature is linearly decayed from 10 to 0.1.

**Self-Evaluated Template Network (SETN)** (Dong & Yang, 2019a). We use the most hyper-parameters as that of DARTS but train it for 250 epochs in total. After training the shared parameters, we select 100 architectures with the highest probabilities (encoded by the learned architecture en-

| Methods | CIFAR-10 Validation Set | | |
| --- | --- | --- | --- |
| | Probability | OSVA (BN with Train) | OSVA (BN with Eval) |
| DARTS-V1 | 0.0779 | 0.0039 | -0.0071 |
| DARTS-V2 | 0.0862 | 0.0355 | 0.0109 |
| SETN | 0.0682 | 0.9049 | 0.0862 |
| GDAS | 0.2714 | 0.8141 | 0.2466 |

Table 7: The correlation between the probability or the one-shot validation accuracy (OSVA) and the ground truth accuracy on the CIFAR-10 validation set. "BN with Train" indicates that, during evaluation, the mean and variance of BN layers are calculated within each mini-batch. "BN with Eval" indicates that we accumulate mean and variance of BN layers in the training set and use these accumulated mean and variance for evaluation. We report the correlation as the average of 3 runs.

coding). We evaluate these 100 selected architectures with the shared parameters. The evaluation procedure for these 100 architectures are the same as RSPS.

**ENAS** (Pham et al., 2018). We use a two layer LSTM as the controller with the hidden size of 32. We use the temperature of 5 and the tanh constant of 2.5 for the sampling logits Following (Pham et al., 2018), we also add the the controller's sample entropy to the reward, weighted by 0.0001. We optimize the controller with Adam using the constant learning rate of 0.001. We optimize the network weights with SGD following the learning rate scheduler as the original paper and the batch size of 128. We did not impose any penalty to a specific operation.

**BOHB** (Falkner et al., 2018). We choose to use BOHB as an HPO algorithm on our *NAS-Bench-201*. We follow (Ying et al., 2019) to set up the hyper-parameters for BOHB. We set the number of samples for the acquisition function to 4, the random fraction to 0%, the minimum-bandwidth to 0.3, the bandwidth factor to 3. We finish the algorithm once the simulated training time reaches the time budgets (12000 seconds).

## C  DISCUSSION FOR NAS WITH PARAMETER SHARING

Parameter sharing (Pham et al., 2018) becomes a common technique to improve the efficiency of differentiable neural architecture search methods (Liu et al., 2019; Dong & Yang, 2019b;a). The shared parameters are shared over millions of architecture candidates. It is almost impossible for the shared parameters to be optimal for all candidates. We hope to evaluate the trained shared parameters quantitatively. Specially, we use DARTS, GDAS, and SETN to optimize the shared parameters and the architecture encoding on CIFAR-10. For each architecture candidate, we can calculate its probability of being a good architecture from the architecture encoding following SETN (Dong & Yang, 2019a). In addition, we can also evaluate a candidate using the shared parameters on the validation set to obtain "the one-shot validation accuracy". It is computationally expensive to evaluate all candidates on the whole validation set. To accelerate this procedure, we evaluate each architecture on a mini-batch with the size of 2048, and use the accuracy on this mini-batch to approximate "the one-shot validation accuracy". Ideally, the architecture ranking sorted by the probability or the one-shot validation accuracy should be similar to the ground truth ranking. We show the correlation between the proxy metric and the ground truth validation accuracy in Table 7. There are several observations: (1) The correlation between the probability (encoded by the architecture encoding) and the ground truth accuracy is low. It suggests that the argmax-based deriving strategy (Liu et al., 2019) can not secure a good architecture. It remains open on how to derive a good architecture after optimizing the shared parameters. (2) The behavior of BN layers is important to one-shot validation accuracy. The accumulated mean and variance from the training set are harmful to one-shot accuracy. Instead, each architecture candidate should re-calculate the mean and variance of the BN layers. (3) GDAS introduced Gumbel-softmax sampling when optimizing the architecture encoding. This strategy leads to a high correlation for the learned probability than that of DARTS. (4) The uniform sampling strategy for training the shared parameters (Dong & Yang, 2019a) can increase the correlation for one-shot accuracy compared to the strategy of the joint optimizing strategy (Dong & Yang, 2019b; Liu et al., 2019).

## D    DETAILED INFORMATION OF *NAS-Bench-201*

In *NAS-Bench-201* (version 1.0), every architecture is trained at least once. To be specific, 6219 architectures are trained once, 1621 architectures are trained twice, 7785 architectures are trained three times with different random seeds. Moreover, we are actively training all architectures with more seeds and will continue updating our *NAS-Bench-201*.

The latency in our *NAS-Bench-201* (version 1.0) is computed by running each model on a single GPU (GeForce GTX 1080 Ti) with a batch size of 256. We report the latency on CIFAR-100 and ImageNet-16-120, and the latency on CIFAR-10 should be similar to CIFAR-10.

**The usage of API.** We provide convenient APIs to access our *NAS-Bench-201*, which can be easily installed via "**pip install nas-bench-201**". Some examples are shown as follows:

```python
from nas_201_api import NASBench201API as API
api = API('NAS-Bench-201-v1_0-e61699.pth')
for i, arch_str in enumerate(api):        # show every architecturre
  print ('{:5d}/{:5d} : {:}'.format(i, len(api), arch_str))
info = api.query_meta_info_by_index(1)  # get metrics of the 1-th arch
res_dict = info.get_metrics('cifar10', 'train') # a dict saving loss/acc
print ('The accuracy is {:.2f}'.format(res_dict['accuracy']))
print ('The loss is {:.2f}'.format(res_dict['loss']))
cos_dict = info.get_comput_costs('cifar100') # a dict saving costs
print ('The flops is {:.2f} M'.format(cos_dict['flops']))
print ('The #parameters is {:.2f} MB'.format(cos_dict['params']))
print ('The latency is {:.3f} s'.format(cos_dict['latency']))
# query the index of a specific architecture from API
arch_index = api.query_index_by_arch('|nor_conv_3x3~0|+|nor_conv_3x3~0|
    avg_pool_3x3~1|+|skip_connect~0|nor_conv_3x3~1|skip_connect~2|')
# get results of each trial for a specific architecture
results = api.query_by_index(arch_index, 'cifar100')
print ('There are {:} trials for this architecture [{:}] on cifar100'.
    format(len(results), api[arch_index]))
```

Please see `https://github.com/D-X-Y/NAS-Bench-201` for more kinds of usages. The benchmark data file for API can be downloaded online from `https://drive.google.com/file/d/1SKW0Cu0u8-gb18zDpaAGi0f74UdXeGKs/view`.

