# OpenReview forum: "NAS-Bench-201: Extending the Scope of Reproducible Neural Architecture Search"
_ICLR.cc/2020/Conference — Accept (Spotlight)_

### Official Review · AnonReviewer1 · 2019-10-18
**Official Blind Review #1**

**Rating:** 8

**Review:**

Summary:

Research into Neural Architecture Search (NAS) has exploded in recent times. But unfortunately the entry barrier into the field is high due to the computational demands of running experiments on even cifar10/100 let alone ImageNet sized datasets. Furthermore there is a reproducibility and fair comparision crisis due to differences in search spaces, training routine hyperparameters, stochasticity in gpu training, etc. This paper proposes a benchmark cell-search space (resnet backbone, 4-node cell space, 5 possible operations) which is algorithm agnostic. They train all possible architectures (15625) in this search space on cifar10/100/Imagenet-16-120 (a reduced version of ImageNet with 120 classes). Thus anyone can now use this pretrained lookup-table to benchmark their search algorithm in seconds on a tiny laptop instead of having to get access to a cluster with hundreds of gpus. By also proposing reference implementations of training architectures the community can use this to fairly benchmark their search algorithms.

The other such benchmark is NASBench-101 which uses a much more expansive search space but by imposing a limit on the number of edges in the cell (to keep the search space manageable with respect to how many of them they have to train) they leave out algorithms which do weight-sharing (ENAS, DARTS, RANDNas) from being able to use their benchmark. This paper alleviates those constraints and thus brings important algorithm classes to their fold.

Comments:

- The paper is very well written. Thanks!

- Minor clarification question: One nice thing of the NASBench paper was the fact that they also reported variance in training with differnt random seeds. I see a line in the 'Metrics' section saying that this is also done but did not find any details on number of trials and whether this was part of the benchmark lookup. I might have missed it somewhere.

- There is another class of search algorithms which grow from small to big cells (if using a cell search space) like EFAS (Efficient Forward Architecture Search by Dey et al and AutoGrow by Wen et al.). Can such algorithms take advantage of this benchmark? I think the answer is yes, because of the 'zeroise' operation but wanted to get the authors' answer.

- Overall I think this is an important contribution to the field and I am assuming that the authors plan to release the benchmark and reference implementations if accepted?

**Experience Assessment:**

I have published one or two papers in this area.

**Review Assessment: Checking Correctness Of Derivations And Theory:**

N/A

**Review Assessment: Checking Correctness Of Experiments:**

I carefully checked the experiments.

**Review Assessment: Thoroughness In Paper Reading:**

N/A

---

> ### Author Response · Authors · 2019-11-09
> **Training details; Supports for algorithms with growing cells; Benchmark Release**
>
> We appreciate your constructive comments and suggestions. Please find our response to each of your questions/comments in the following.
>
> Q1. More details on the number of trials and whether this was part of the benchmark lookup.
>
> R1. In the current version of our AA-NAS-Bench, every architecture is trained at least once. To be specific, 7433 architectures are trained once, 782 architectures are trained twice, 7410 architectures are trained three times with different random seeds. Our API supports returning the metrics of a specific trial. Moreover, we are actively training all architectures with more seeds and will continue updating our AA-NAS-Bench. We have clarified this information in the footnote on Page 4. We plan to finish the training of all architectures for 3 trials in 4 months.
>
> Q2. Can the searching algorithms which grow from small to big cells take the advantages of this benchmark?
>
> R2. Yes, they can take advantages of this benchmark, because each small cell is equivalent to one big cell by adding some “skip-connect” and “zeroize” operations. Please see the following example.
> A small cell with 3 nodes:
> node-1 -> node-2: 3x3conv
> node-1 -> node-3: 3x3conv
> node-2 -> node-3: 3x3conv
>
> The corresponding big cell with 4 nodes:
> node-1 -> node-2: 3x3conv
> node-1 -> node-3: 3x3conv
> node-2 -> node-3: 3x3conv
> node-1 -> node-4: zeroize
> node-2 -> node-4: zeroize
> node-3 -> node-4: skip-connect
>
> Therefore, our AA-NAS-Bench can also provide the metrics for all small cells, and benefit to searching algorithms that grow from small to big cells, e.g., EFAS and AutoGrow.
>
> Q3. Release the benchmark and reference implementations.
>
> R3. Of course. We will release all source codes for training each architecture candidate and baseline searching algorithms during the rebuttal period. We would also provide convenient APIs to access our benchmark.

---

> > ### Comment · AnonReviewer1 · 2019-11-11
> > **Agreed!**
> >
> > Thanks for the confirmation! Looking forward to the full benchmark.

---

> > > ### Author Response · Authors · 2019-11-14
> > > **Upload codes and data**
> > >
> > > Thanks for your recognition. We just uploaded all codes and data to the anonymous links as follows:
> > >
> > > 1. Codes at https://github.com/D-X-Y/NAS-Projects include
> > > - instruction on how to re-generate our dataset
> > > - usages of 10 re-implemented NAS algorithms
> > > - instruction on how to use our API
> > >
> > > 2. The data for API is at https://drive.google.com/file/d/1SKW0Cu0u8-gb18zDpaAGi0f74UdXeGKs/view

---

> > > > ### Comment · AnonReviewer1 · 2019-11-15
> > > > **Acknowledged!**
> > > >
> > > > Thanks for the links to code and data.

---

### Official Review · AnonReviewer3 · 2019-10-23
**Official Blind Review #3**

**Rating:** 8

**Review:**

--- Updated during response period ---

Authors successfully answers all my questions. I revise my rating to Accept.


-----

Summary:

This paper proposes another benchmark dataset for neural architecture search. The idea is following the NASBench-101 dataset, that in a given search space, densely sampled all existing architectures and train each of them on three tasks for multiple times, and using the obtained metrics as a tool to evaluate an arbitrary neural architecture search algorithm. The paper also presents comprehensive reports on the statistics, revealing a strong performance correlation between tasks, and evaluate some baseline NAS algorithms.

I think this paper will be valuable to the research community for these reasons: (1) the dataset contains a more geologically complex search space comparing to the original NASBench-101, whose search space is restrained in certain ways; (2) released metrics include more meaningful information rather than single point value in NASbench; (3) it uses 3 datasets rather than 1.
My major concerns, which I will detail later, is the phrasing "algorithm-agnostic" does not truly reflect the difference between their approach and NASBench-101, and about the architecture search space design details.

Altogether, I think even the technical novelty is incremental, the work is not trivial considering the computational cost. I am willing to improve my score if my concerns are addressed during the rebuttal period. Nevertheless, this dataset is a strong subsidy of existing NASBench-101 and can benefit the research community and serves as an important baseline to evaluate a NAS algorithm.


Strength

+ Clear motivation to use an operation-on-the-edge search space that is widely used in NAS domain.
+ Extensive experiments on evaluating 15K architectures over 3 datasets
+ Detailed statistics on the search space
+ Good baseline experiments comparison

Main concerns about this dataset:

- Comparing to NASBench-101 in terms of "Algorithm Agnostic", it is in a "more-or-less" game but not a "yes-or-no" one, so that AA-NAS-Bench does not seem appropriate. In my perspective, this dataset has not shown significant differences for the following reasons.

1. With proper adaptation, both NASBench-101 and the one in this paper are "algorithm agnostic". For example, original ENAS is training a reinforcement learning sampler that learns to predict a string with encoding [id1, op1, id2, op2] for each node, where id1, id2 is the IDs of the previous node to connect, op1, op2 is the operation choice for each edge. Since NASBench has operation on output node, one could simply make RL sampler to predict [id1, id2, op1], or another string encoding that suits the search space better. In my perspective, Ying et al. mentioned that many NAS algorithms cannot be directly evaluated on NASBench-101 are because the search space is different, but it does not mean using NASBench-101 is impossible. On the other hand, for some other state-of-the-art algorithms, like Proxyless-NAS on ImageNet, the search space is also different from the one proposed in this paper, but likewise, it does not indicate evaluating Proxyless-NAS on this dataset is impossible.

2. NASBench-101 does impose constraints on maximum edge number equals to 9 with 7 nodes in their space and results in 423K architectures. However, this constraint is no longer applied if you reduce the number of node to 6 (i.e. all possible architectures can be sampled), yet it still contains around 64K architectures, which is more than 15K in the proposed dataset. In this perspective, NASBench is a larger dataset and "algorithm agnostic".

To summarize, I acknowledge the paper's contribution is using an operation-on-the-edge search space that is widely used in previous NAS algorithms while NASBench-101 is using operation-on-the-node space. However, it only makes the proposed dataset "more algorithm agnostic" with less effort, and it does not make the previous NASBench-101 "not" algorithm agnostic. If using the current name AA-NAS-Bench, I think it is not fair for the NASBench-101, specifically they are 4 times larger after removing the edge number constraints.

- Questions about architecture space design
1. Why using average pooling instead of max pooling?
2. How do you compute the total architecture number 15,625 in Table 3? In your setting, with the number of node V=4 densely connected DAG, it should have 6 edges as depicted in Figure 1, and each edge has 5 possible operations, i.e. total number = 6^5 = 7776. I am confused about this point, could author comment more on this number?
3. Is there any topologically equal architectures in this space? For example, let's name the node 1,2,3,4, and the following two architectures should be the same since input edges are summed before passed to the next node. I listed **non-zeroed** edge as, id1->id2: op

Architecture 1:
1->2: conv3x3
2->4: skip
1->3: conv1x1
3->4: skip

Architecture 2:
1->2: conv1x1
2->4: skip
1->3: conv3x3
3->4: skip

If the pruning is not effectively conducted, my worry is the actual number of architectures is smaller.

Minor comments:
1. DARTS results on the are quite poor as mentioned in this paper that, DARTS will eventually converge to an architecture with all skip connection. However, it could be a simple fix, by tracking the architecture evolution during the search and report the best like early-stopping. Will this improve DARTS results?

2. Since ENAS is the first work using parameter sharing on the NAS problem, could the author add it to the baseline?

3. In table 4, what is the average (94.37 for CIFAR-10) mean in the "optimal" column? Is this the mean performance of all architectures? If so, it is quite strange to see all the baselines are selecting architectures worse than the average performance. Or it is the best architecture performance as indicated in the caption? This "average" column for "optimal" seems confusing.

4. The dynamic ranking of architecture in Figure 5 is very interesting. Architecture ranking seems stable after the 190th epoch. Could the author provide another visualization, showing when stabilization happens in between epoch number 150 and 190?

5. Figure 4, correlation matrix for top 4743 architectures are significantly lower than the full and 1387 ones, is this possible because of repetitive architectures in the space are not pruned? And, what is the reason for number 4743 and 1387?

6. ResNet (star in Figure 2) seems to perform very well. Does this indicates the proposed search space is not much meaningful, considering there are only 1~2% for NAS to improve?

**Experience Assessment:**

I have published one or two papers in this area.

**Review Assessment: Checking Correctness Of Derivations And Theory:**

N/A

**Review Assessment: Checking Correctness Of Experiments:**

I carefully checked the experiments.

**Review Assessment: Thoroughness In Paper Reading:**

I read the paper thoroughly.

---

> ### Author Response · Authors · 2019-11-09
> **More comparison with NASBench-101; Clarification on our benchmark.**
>
> Thank you for your constructive and detailed review. We have updated the paper according to your comments and suggestions. Detailed responses are shown in a point-to-point manner below.
>
> Q1. More detailed discussion and comparison with NASBench-101.
> R1. We agree with the reviewer’s statement: with some modification to both NASBench-101 (a reduced one) and NAS algorithms, most algorithms could also be evaluated on the modified NASBench-101. To the best of our knowledge, such modification is non-trivial and might be beyond the scope of the original NASBench-101 paper. Also, the modifications might need extra tedious effort, which is no longer convenient to use and against the main motivation of the benchmark.
>
> A subset of NASBench-101 with all possible architectures included needs to have 4 or fewer nodes, which sum to only less than 500 architectures. This is because a complete DAG with n nodes has n*(n-1)/2 edges and NASBench-101 limits the maximum number of edges to 9, therefore, the number of nodes (n) should be <= 4.
>
> Q2. Why using average pooling instead of max pooling?
> R2. It is inspired by the typical architectures, such as ResNet and ResNeXt (https://github.com/facebookresearch/ResNeXt/blob/master/models/resnext.lua#L38), which use average pooling in their residual blocks.
>
> Q3. How do you compute the total architecture number 15,625 in Table 3?
> R3. There are 6 edges when we use the number of node V=4. Each edge has 5 possible operations. Therefore, the total number is 5^6 = 15625.
>
> Q4. Are there any topologically equal architectures in this space? Is the actual number of architectures smaller?
> R4. Yes, the number of unique architectures is 12751.
>
> Q5. How about early stopping on DARTS to avoid finding the architecture with all skip connection.
> R5. We follow the original training strategy in the DARTS paper. Even if the early stopping may improve the performance, it is not the focus of this paper.
>
> Q6. Add ENAS as a baseline NAS algorithm.
> R6. Thanks for this suggestion. We have included ENAS in Table 4.
>
> Q7. Clarify the “optimal” column in Table 4.
> R7. We average the accuracy results of all trials for each architecture. The “optimal” means the highest mean accuracy. We have revised Table 4 to clarify it.
>
> Q8. Could the author provide another visualization, showing when stabilization happens in between epoch number 150 and 190?
> R8. We have made a video to show the ranking over training epochs. Please see the video at https://drive.google.com/open?id=1rp58l5FM-3Q-S7tPSYX003BVekWkj8X7
>
> Q9. Figure 4, correlation matrix for top 4743 architectures are significantly lower than the full and 1387 ones, is this possible because of repetitive architectures in the space are not pruned? And, what is the reason for the number 4743 and 1387?
> R9. Thanks for pointing out this problem. Figure 4b and 4c should be exchanged. We have updated it. The number of architectures is derived by the number of top architectures with accuracy > 92% (4743) and 93% (1387).
>
> Q10. ResNet (star in Figure 2) seems to perform very well. Does this indicate the proposed search space is not much meaningful, considering there are only 1~2% for NAS to improve?
> R10. This is also true for NAS-Bench-101: ResNet is competitive and most NAS algorithms just find a worse architecture than ResNet. As shown in Table 4, the best architecture found by 10 NAS algorithms is still far from the best architecture in the search space (> 1% on CIFAR-10 and > 3% on CIFAR-100 and ImageNet-16-120). Also, the stability of the NAS algorithms should also be considered since the evolution-based and RL-based methods usually suffer from high variance. In Section 6, we included the rules to use the benchmark to avoid boosting performance using priors, e.g., hard-code rules.

---

> > ### Comment · AnonReviewer3 · 2019-11-14
> > **Rebuttal resolves the majority of my concerns**
> >
> > Author responses address most of my concerns, except they did not respond to the naming of their dataset.

---

> > > ### Author Response · Authors · 2019-11-15
> > > **Naming**
> > >
> > > Thanks for your comments.
> > > As replied in Q1 above, since (1) non-trivial modifications are required to evaluate all NAS algorithms on NAS-Bench-101 and (2) a subset of NASBench-101 which includes all possible architectures have 4 or fewer nodes, which sum to only less than 60 unique models, we believe NAS-Bench-101 is not qualified as an algorithm-agnostic NAS benchmark. This limitation has already been mentioned in the original paper: "NAS algorithms based on weight sharing (Pham et al., 2018; Liu et al., 2018b) or network morphisms (Cai et al., 2018; Elsken et al., 2018) cannot be directly evaluated on the dataset, so we did not include them". This is the main motivation for our benchmark.
> > > We welcome the discussion about naming. Do you have some candidates for the name of our benchmark?

---

### Official Review · AnonReviewer2 · 2019-10-23
**Official Blind Review #2**

**Rating:** 8

**Review:**

Edit after rebuttals: I have read all other reviews and rebuttals and maintain my assessment.
----
Summary: Comparison of neural architecture search algorithms is hindered by the lack of a common measurement procedure. This paper describes a publicly available benchmark on which most recent types of NAS algorithms can be evaluated. It does so by exhaustive calculation of performance metrics on the full combinatorial space of select architectures, on two select datasets. NAS algorithms can then perform search without having to perform evaluation on each node, which shrinks the computational cost of experimentation and benchmarking drastically.

I recommend acceptance, as the resource described in the paper has been created thoughtfully and is useful to the research community, as well as to users of NAS algorithms. The paper is clear about restrictions too, which doesn't hurt.

The technical details are laid out clearly especially in sec 2.1. It would be interesting to know the computational cost of producing the data. It is useful in practice to have access to different metrics (validation, training and test) for each node, as well as extra diagnostic information.

The usefulness of the resources hinges on a few elements, which make its strength and also weakness:
- choice of tasks and datasets
- choice of skeleton architecture, fig 1
- choice of hyperparameters, sec 2.3 (I note there is no regularisation, as discussed in the paper)
All of these seem reasonable to me. It is clearly a limitation that hyperparameter search is infeasible to conduct in parallel with architecture search, as pointed out sec 6.

The principal competitor NAS-Bench-101 is only applicable to specific NAS algorithms, which evidences the need for the present resource. The discussion and comparison in sec3 is fair.

The discussion of weaknesses, such as possible overfitting patterns, or technical choices, is balanced.

# Minor
English proofreading is required.
- Maybe you can attempt a pun on Ananas in the naming?
- I'm not sure "fairness" as in the abstract is the exact core problem; I would call this comparability.
- sec2 head: "side information", I usggest diagnostic information
- sec2.2 "and etc" is a redundant: etc stands for "and the others"
- sec2.4 almost involves almost; target on computation cost; stabability
- sec 4: has impacts on, parameters keeps the same -> stays, which serves as testing -> to test
- sec6 tricky ways-> insidious?

**Experience Assessment:**

I do not know much about this area.

**Review Assessment: Checking Correctness Of Derivations And Theory:**

N/A

**Review Assessment: Checking Correctness Of Experiments:**

I assessed the sensibility of the experiments.

**Review Assessment: Thoroughness In Paper Reading:**

I read the paper at least twice and used my best judgement in assessing the paper.

---

> ### Author Response · Authors · 2019-11-09
> **Minor English Writing Problem**
>
> We appreciate your recognition of our paper and valuable comments regarding writing. Please find our response to each of your questions/comments in the following.
>
> - Try to attempt a pun on Ananas in the naming?
> We are brainstorming this problem. Do you have some suggestions?
>
> - I'm not sure "fairness" as in the abstract is the exact core problem; I would call this comparability.
> Nice correction. We have revised the paper according to your suggestion.
>
> - sec2 head: "side information", I suggest diagnostic information
> Thanks for this constructive suggestion. We have replaced all “side information” with “diagnostic information”.
>
> - sec2.2 "and etc" is redundant: etc stands for "and the others"
> We have revised the sentence to “The test set is to evaluate the performance of each searching algorithm by comparing the indicators (e.g., accuracy, model size, speed) of their selected architectures.”.
>
> - sec2.4 almost involves almost; target on computation cost; stabability
> Thanks for your comments. We have revised the sentences as:
> (1) Collecting these statistics almost involves no extra computation cost
> (2) Algorithms that target on searching architectures with computational constraints, such as models on edge devices, can use these metrics directly in their algorithm designs without extra calculations.
> (2) the stability
>
> - sec 4: has impacts on, parameters keeps the same -> stays, which serves as testing -> to test
> Thanks for your comments. We have revised the sentences as:
> (1) Results show that a different number of parameters will affect the performance of the architectures, which indicates that the choices of operations are essential in NAS.
> (2) We also observe that the performance of the architecture can vary even when the number of parameters stays the same.
> (3) The performance of the architectures shows a generally consistent ranking over the three datasets with slightly different variance, which serves to test the generality of the searching algorithm.
>
> - sec6 tricky ways-> insidious?
> Good suggestion. We have revised “tricky” by “insidious”.

---

### Comment · Area_Chair2 · 2019-11-09
**Code availability**

The paper states "All code, data, and architecture information are publicly available."

Where is it available? Please post an anonymized version of this during the rebuttal phase. This is absolutely crucial for a paper proposing a new benchmark.

---

> ### Author Response · Authors · 2019-11-14
> **Full Code and Data Release**
>
> Dear AC,
>
> Thanks for your comment. We have uploaded the codes and data to anonymous links as follows:
>
> 1. Codes at https://anonymous.4open.science/repository/9aa95a13-7e6a-48ed-9c77-4ac9111f7ae9/README.md include
> - instruction on how to re-generate our dataset
> - usages of 10 re-implemented NAS algorithms
> - instruction on how to use our API
>
> 2. The dataset is at https://drive.google.com/open?id=1qEsEiGnr4HhOoU_2s_z4zRHye7C5LkTi
>
> Best regards,
> Authors of AA-NAS-Benchmark

---

### Author Response · Authors · 2019-11-14
**Update the manuscript**

We thank the AC and all reviewers for their constructive comments. We have updated a revised version of the paper and would like to highlight the changes as follows:

1. We add publicly available codes for reproducing the proposed AA NAS benchmark and all 10 NAS algorithms.

2. We add instructions on how to use our API in the appendix.

3. We add more discussion w.r.t. the applicability of NAS-Bench-101.

---

### Comment · Area_Chair2 · 2019-11-15
**Area Chair Comments 1/2**

Dear authors,

I just read this paper myself, and I have several remarks to which I would like to give you an opportunity to reply.
(I know time in the rebuttal period is short now, and I don't expect a full reply, but it's still better for you this way than if I only raise these points in the private discussion with the reviewers after the rebuttal period is over.)

1. I would like to bring to your attention that there is another parallel ICLR submission called NAS-Bench-1Shot1 (https://openreview.net/forum?id=SJx9ngStPH) that actually shows how to use the original NAS-Bench-101 to benchmark weight sharing algorithms. They reformulate NAS-Bench-101 by mapping nodes to edges and extract 3 different subspaces with 6k, 29k, and 360k architectures, respectively, from the NAS-Bench-101 space that are directly compatible with modern weight-sharing methods, such as ENAS and DARTS.
Of course, this parallel work does not reduce the novelty of yours. However, in light of it, I would like to ask you to rephrase some of your claims about the impossibility of using NAS-Bench-101 for weight-sharing algorithms; it *can* be applied, but it cannot be applied *directly* (the latter was also the wording in the NAS-Bench-101 paper).

2. The paper follows the NAS-Bench-101 paper in motivation and in most design decisions, and makes some (useful and well-taken, but technically incremental) modifications.
Nevertheless, the introduction does not even mention NAS-Bench-101, but rather leaves the unknowing reader with the impression that this is the first paper introducing such a NAS benchmark.
It even states "The AA-NAS-Bench has shown its value in the field of NAS research", which is incorrect; I do believe that it *will* show its value, but so far the only benchmark that *has* shown
its value is NAS-Bench-101 (which indeed has been used by many groups).
I therefore believe that it would be much preferable to state clearly in the introduction that NAS-Bench-101 already exists, and list the useful extensions made here (e.g., in bullet point form):
- operations in the edges to enable weight sharing methods
- multiple datasets
- extra statistics
Likewise, abstract & conclusion should point out what's new compared to NAS-Bench-101. Section 3 is too late to first mention NAS-Bench-101.

3. A minor point: NAS-Bench-101 also performed runs with a shorter training budget in order to obtain networks for which cosine annealing had converged; the performance of these converged networks is much more likely to correlate highly with the performance after a larger number of iterations than just taking an earlier point of a single cosine annealing trajectory. Since these high correlations are required for bandit-based algorithms (such as Hyperband and BOHB) to perform well, your benchmark may be argued to not support these well and thus be *less* agnostic to the choice of algorithms than the original NAS-Bench-101.

4. Concerning the name, I believe the points made by AnonReviewer3 and the points above clearly speak against "algorithm-agnostic" as part of the name.
Since the benchmark is very similar to NAS-Bench-101, with some things changed, I believe a very natural name would be NAS-Bench-102. (We should reserve NAS-Bench-201, etc for much larger search spaces.) A corresponding paper title could be "NAS-Bench-102: Extending the Scope of Reproducible Neural Architecture Search" or alike.

5. A minor point: technically, this benchmark is not the first that evaluates on multiple datasets. There are the NAS-HPO-Bench Datasets, which the NAS-Bench-101 used to decide about the strategy for fixing hyperparameters. These NAS-HPO-Bench Datasets (https://arxiv.org/abs/1905.04970) evaluate 62208 configurations in the joint NAS+HPO space of a simple feed-forward network, each of them on 3 datasets. So, technically this has been done before, but the architectures in that paper are so simple that this does not take away from the novelty of this contribution of the paper.

---

> ### Comment · Area_Chair2 · 2019-11-15
> **Area Chair Comments 2/2**
>
> 6. Weight sharing algorithms will actually *not* be super cheap to evaluate, even with your benchmark; you couldn't just do this quickly on a laptop (as one of the reviewers thought). The only part of the computation that can be saved is that for their final evaluation step, but the search phase still needs to be carried out manually, and this also takes several GPU hours per run. I assume this is also the reason that you only report 3 runs in your Table 4, rather than the 500 (!) runs that the NAS-Bench-101 reported. Please emphasize this very clearly. Otherwise, this would be misleading, and reviewers of future NAS papers using the dataset would likely think that runs on the dataset should be super-fast and complain that authors don't carry out more runs. This is particularly important since this issue occurs precisely for the weight-sharing algorithms you want to support with this new benchmark. Please include the time taken for each of the algorithms you report (broken up into time the NAS algorithm used internally for the search phase and simulated time for results read from your results table).
>
> 7. The comparison of NAS algorithms in Table 4 is poor for a paper presenting a new benchmark.
>  7.1 Why did you only perform 3 runs? The whole point is that your benchmark should be cheap to evaluate and not require a lot of compute. The NAS-Bench-101 paper reported 500 runs per algorithm. At least for the algorithms that only query the table and don't have to train weight-sharing models, please carry out more runs and report statistics.
>  7.2 Why do you not plot performance as a function of time (simulated time, pretending you actually evaluated the architectures being queried)? The NAS-Bench-101 paper already did this (see Figure 7 (left) there), and you already have the table to evaluate all the found architectures in zero time, so only reporting final performance of the methods is a big step back. Also see the recent checklist for best practices in NAS evaluation, best practice 7: https://arxiv.org/abs/1909.02453
>  7.3 Even worse, the methods apparently did not receive the same amount of time (sum of actual compute time used + simulated time for querying the table), so you are comparing apples and oranges. That is not OK for a paper proposing a benchmark to the community (everybody else would follow suit and do this wrongly; this would be a step back rather than a step forwards for empirical evaluations in the community). If you really want to put results into a table (rather than in a Figure like suggested above in 7.2), then you should ensure that all methods were allowed the same amount of time (also see best practice 13 in the checklist above).
>  7.4 For bandit-based algorithms, if you evaluate with a lower budget (e.g., 10 epochs rather than 100), you should only count the simulated time as a fraction of the time stored in your table (in the example 10/100 of that time). It is not the number of table queries that counts, but the sum of the simulated time for the entries queried.
>
> 8. A minor point: in Table 3, the #architectures of NAS-Bench is 423k after treating isomorphisms, but you include isomorphic graphs in your 15k count; so this should be 12751, or the number for NAS-Bench-101 should be changed.
>
> 9. Having made all of the points above, I would like to emphasize that new benchmarks are dearly needed for NAS; this was, for example, also identified as one of the most urgent action items in the panel of the AutoML workshop at ICML 2019. I therefore strongly welcome work on this problem. My comments above are critical, and, if left unaddressed, may lead to some reviewers reconsidering their assessment of the paper, but I very much hope that you will respond to this message to alleviate worries about these points and allow us to accept your paper. Due to the short time left in the rebuttal period, I hope that you can adapt the easy-to-address points in the paper, and for the others provide a reply and promise according adaptations in the paper.
>
> Best,
> Your Area Chair

---

> > ### Author Response · Authors · 2019-11-15
> > **[Thanks for your constructive comments] Name change; Rephrase claims; More analysis; More experimental results.**
> >
> > Many thanks for your constructive comments and suggestions.
> > Even if the replying period is short (< 10 hours to reply your comments and it was close to sleep time when we received it), we try our best to address most of your points in the paper and this response, and we promise to continue revising the rest in our paper regarding the writing and experimental results.
> >
> > Q1-Q2: Mention NAS-Bench-101 in the abstract & introduction. Rephrase some claims of NAS-Bench-101.
> > R1. Thanks for this suggestion. We have revised the manuscript accordingly.
> >
> > Q3: Limitation for bandit-based algorithms.
> > R3. Agreed. We have added a paragraph to discuss this limitation (Sec 6 in Page 10). It remains an open question whether (1) training a network 10 epochs with converged cosine annealing can provide a higher correlation than (2) training a network 10 with unconverged cosine annealing or not. We promise to add experiments to compare the correlation of these two strategies in our revision.
> >
> > Q4: Suggestions for the name.
> > R4. Thanks for the suggestion and we have revised it.
> >
> > Q5: Not the first benchmark that evaluates on multiple datasets.
> > R5. We have revised the paper accordingly (Page 2).
> >
> > Q6: Emphasize that weight sharing algorithms still require several GPU hours. Include the time taken for each NAS algorithm.
> > R6. Thanks for the suggestion. We have added one paragraph and a table in Sec. 5 to emphasize this problem. We need some time to fairly compare the time cost for each NAS algorithm and promise to include it in our revision.
> >
> > Q7.1 Results of 500 runs for algorithms without weight sharing.
> > R7.1. Thanks for the suggestion. We have included new results with 500 runs for REA/REINFORCE/RANDOM/BOHB in Table 5. We have drawn a new figure to show all the results of these 500 runs in Figure 6. We will re-arrange the latex layout later.
> >
> > Q7.2 Plot performance as a function of time.
> > R7.2 Thanks for the suggestion. We are modifying the codes and running the suggested experiments. We promise to include these results in our revision.
> >
> > Q7.3 Using time budget instead of "number of networks".
> > R7.3. Nice suggestion. We promise to revise the experiments (Table 5 and Figure 6) for REA/REINFORCE/BOHB/RANDOM using the time budge.
> >
> > Q8: Using #unique architectures in Table 3.
> > R8. Thanks for the suggestion, and we have revised Table 3.

---

### Public Comment · ~Chris_Ying2 · 2020-01-12
**Naming of the benchmark**

As the authors of the NAS-Bench-101 benchmark we have discussed the naming of this follow-up benchmark with the authors of this paper and with the AC, and we have unanimously concluded that NAS-Bench-201 would be a slightly better name, as it may lead to less confusion (it is neither a subset nor a superset of NAS-Bench-101) and this also allows to use “NAS-Bench-20x” for minor updates and consistent names NAS-Bench-301, NAS-Bench-401, etc for future benchmarks.

---

> ### Author Response · Authors · 2020-01-12
> **Agreed**
>
> Thanks a lot for this constructive suggestion.
> I agreed with your point and it makes sense to me. I will revise the manuscript soon.

---

### Public Comment · ~Philipp_Jamscikov1 · 2020-01-21
**Description and Naming of Train/Val/Test Splits in Paper and Source Code**

Dear authors,

I am a graduate student relatively new to the field and would have preferably opened an issue on Github, which was not possible at the point in time.  As arguably any NAS/AutoMLBench targets persons with a similar non-expert background, I thought you may benefit from this user-oriented review.

For datasets other than CIFAR-10, I find it difficult to link the train/val/test splits described in the paper to the corresponding numbers being reported in the source code (NAS-Bench-201-v1_0-e61699.pth).

Conceptually, for every random seed, architecture, and dataset, I would expect most importantly something similar to this being reported:
- training performance for every of 200 epochs
- performance on the validation set for every of 200 epochs - supervision signal during training
- performance on unseen test data: select model configuration with lowest validation loss and predict (once and at the very end) the test set

As explained in the paper, one could expect these result for all datasets expect for one case of CIFAR-10, where it is being trained on train+val, and with the default test set serving as a supervision signal.

When looking in the source code, I generally would proceed as follows: create an ArchResults object, query it for a specific dataset and seed,  and look at the “eval_acc1es” dictionary which contains the validation performance for 200 epochs and as a last entry the test performance on the test set.

E.g., for ’CIFAR-10-valid’ (which is the not-so-obvious identifier for dataset for CIFAR-10 being trained on train only) we get a performance for ’x-valid@0' - 'x-valid@199’, and a single more entry in the diction 'ori-test@199’. This is what I would have expected and corresponds to what I described above.

However, for the datasets CIFAR-100  and ImageNet16-120, I struggle to understand the corresponding “eval_acc1es” dictionary.
In both cases, we now have  200 epochs of ’ori-test@...’ being reported, followed by two entries: 'x-valid@199' and
 'x-test@199’).
In section 2.2 of the paper, you describe for CIFAR-100 that you split the original test set into a new validation set and test set, which confuses me here even more, as I would expect two and not three metrics being reported in the “eval_acc1es” dictionary. This also holds w.r.t. to ImageNet16-120 and its description in the paper.
Section 2.3, and especially Table 2 in the paper do not help to clarify these points. From the table I might (misleadingly?) conclude that the NASBench is run twice for ImageNet-16-120 and CIFAR-100, one time with the validation set, the other time with the test set serving as a supervision signal during training.

I hope you may find this direct user feedback of your API somewhat useful and thank you for the effort you’ve put so far into the project.

---

> ### Author Response · Authors · 2020-01-21
> **The usage of API**
>
> Thanks for your suggestions.
>
> I should write the README in more detail and I will update the readme according to your suggestion :) It is welcome to open an issue on Github, usually, I will reply in 24 hours.
>
> In short, please try the get_more_info function in the API and you won't care about "ori-" or "x-"..
> It takes args (index, dataset, iepoch=None, use_12epochs_result=False, is_random=True).
> The index is the architecture index from 0-15624.
> The dataset indicates the name of dataset, 'cifar10-valid' indicates training on the train set of CIFAR-10.
> 'cifar10' indicates training on the train+valid set of CIFAR-10.
> 'cifar100' indicates training on the training set of CIFAR-100.
> 'ImageNet-16-120' indicates training on the training set of ImageNet-16-120.
> It will return a dict with the key of 'train-loss' / 'train-accuracy' / 'train-per-time' (per-epoch-time) / 'train-all-time' / 'valid-loss'.
>
> Noticed that, sometimes, the value of a key might be None or the key is not available. In this case, for some of them, we can find an alternative way to calculate it, but for some of them, unfortunately, it is not available at the moment. For details, would you mind to open a GitHub issue? The NAS-Bench-201 goes through several changes, and I need to merge the trained data of different internal benchmark versions, which can yield some confusing names. Sorry for the confusion.

---

### Decision · Program_Chairs · 2019-12-19

**Decision:**

Accept (Spotlight)

**Comment:**

This paper presents a new benchmark for architecture search. Reviewers put this paper in the top tier. I encourage the authors to also cite https://openreview.net/forum?id=SJx9ngStPH in their final version.